# Forest Volatile Organic Compounds and Their Effects on Human Health: A State-of-the-Art Review

**DOI:** 10.3390/ijerph17186506

**Published:** 2020-09-07

**Authors:** Michele Antonelli, Davide Donelli, Grazia Barbieri, Marco Valussi, Valentina Maggini, Fabio Firenzuoli

**Affiliations:** 1Terme di Monticelli, 43022 Monticelli Terme PR, Italy; 2Institute of Public Health, University of Parma, 43125 Parma PR, Italy; 3CERFIT, Careggi University Hospital, 50139 Firenze FI, Italy; donelli.davide@gmail.com (D.D.); valentina.maggini@unifi.it (V.M.); fabio.firenzuoli@unifi.it (F.F.); 4AUSL-IRCCS Reggio Emilia, 42122 Reggio Emilia RE, Italy; 5Binini Partners S.r.l. Engineering and Architecture, 42121 Reggio Emilia RE, Italy; grazia.barbieri.1994@gmail.com; 6European Herbal and Traditional Medicine Practitioners Association (EHTPA), Norwich NR3 1HG, UK; marco@infoerbe.it

**Keywords:** biogenic volatile organic compounds, phytoncides, pinene, limonene, forest, public health, preventive medicine, review

## Abstract

The aim of this research work is to analyze the chemistry and diversity of forest VOCs (volatile organic compounds) and to outline their evidence-based effects on health. This research work was designed as a narrative overview of the scientific literature. Inhaling forest VOCs like limonene and pinene can result in useful antioxidant and anti-inflammatory effects on the airways, and the pharmacological activity of some terpenes absorbed through inhalation may be also beneficial to promote brain functions by decreasing mental fatigue, inducing relaxation, and improving cognitive performance and mood. The tree composition can markedly influence the concentration of specific VOCs in the forest air, which also exhibits cyclic diurnal variations. Moreover, beneficial psychological and physiological effects of visiting a forest cannot be solely attributed to VOC inhalation but are due to a global and integrated stimulation of the five senses, induced by all specific characteristics of the natural environment, with the visual component probably playing a fundamental role in the overall effect. Globally, these findings can have useful implications for individual wellbeing, public health, and landscape design. Further clinical and environmental studies are advised, since the majority of the existing evidence is derived from laboratory findings.

## 1. Introduction

### 1.1. Forest Volatile Organic Compounds (VOCs): Background and Natural Functions

The first systematic analysis of the importance and role of volatile organic compounds (VOCs) emitted by plants in the atmosphere above forests was probably performed from the perspective of atmospheric chemists who were interested in the effects of VOCs on atmospheric composition and climate. In two seminal articles [1,2], the Dutch scientist Fritz Went and his colleagues hypothesized that the blue haze observed above many forests was caused at least partially by volatile organic compounds emitted by plants (forest VOCs) and went on to estimate the impact that biogenic emissions have on the global balance of organic compounds released into the atmosphere, compared to anthropogenic production. The first estimate of global terpene emissions was 175 millions of tons C/year [1], but more recent evaluations state that worldwide biogenic VOC (BVOC) emissions into the atmosphere could amount to around 1 billion of tons C/year [3]. Forests seem to be the greatest BVOC emitters. Specifically, tropical trees appear to be responsible for around 80% of terpenoid emissions and for around 50% of other BVOC emissions, while other tree species seem to contribute to around 10% of total BVOC emissions [3]. BVOCs form 80% to 90% of total VOC emissions in the atmosphere each year [4,5].

Some forest VOCs are also defined as “phytoncides” and this term, first introduced in the last century by the Russian biologist Boris Petrovich Tokin, derives from the combination of two words: “phyton”, which refers to plants and botany in ancient Greek, plus the suffix “cide”, which indicates “killing” or “exterminating”, thus underscoring the antimicrobial and insecticidal activity of these substances [6,7]. However, the term “phytoncide” may be misleading because, due its broad definition and etymology, it can indicate either plant-derived volatile substances with antiparasitic properties or any (volatile and nonvolatile) antimicrobial/insecticidal compound released by plants, any volatile VOC, or, in some cases, essential oils obtained from aromatic woods. In order to avoid potential misunderstandings, in this research work, which is focused on biogenic volatile substances found in forest air, the term “forest VOCs” will be preferred over the less specific “phytoncide”, and BVOCs will be defined as any organic compound, except carbon dioxide and monoxide, emitted by plants and having vapor pressure high enough to be vaporized in relevant amounts. Usually, the definition excludes dimethyl sulfide and methane because it is still debated as to whether they are produced by terrestrial plants [8].

More than 1000 different BVOCs are released from plant flowers, vegetative parts, or roots [9]. These substances are largely lipophilic products with molecular masses under 300 Da, and the vast majority are isoprenoids, including hemiterpenes (C5H8) such as isoprene, monoterpenes (C10H16), irregular acyclic homoterpenes (C11H18 or C16H26), and sesquiterpenes (C15H24), both as hydrocarbons and as oxygenated compounds; there is also a number of low-molecular-weight molecules, especially C1 and C2 oxygenated compounds, such as methanol and acetaldehyde, and C6H10 fatty acid derivatives, including lipoxygenase pathway products such as green leaf volatiles (henceforth GLVs), plus benzenoids and phenylpropanoids [5,9,10,11,12,13].

Many BVOCs are common to phylogenetically distant plant species [14]. For instance, only some plant species, even belonging to taxonomically distant taxa (like *Bryophyta* and *Quercus*), emit isoprene, while in close taxa, there can be both emitters (*Quercus* spp.) and non-emitters (*Acer* spp.) [11], thus suggesting that metabolic pathways related to their production have been highly preserved during evolution [14].

From the perspective of plant physiology, BVOCs can be divided into constitutive and inducible compounds. All plants can potentially synthesize volatile isoprenoids if triggered by abiotic and biotic stress factors, but only some species can do this constitutively [15].**Constitutive forest VOCs** are already synthesized in the organism and either stored in specialized structures or constantly emitted without any form of storage [16]. They are emitted at baseline levels and are mainly made of terpenoids, plus shikimate derivatives and polyketides [5].**Inducible forest VOCs** (herbivore-induced plant volatiles; henceforth HIPVs) are compounds whose synthesis is increased or initiated de-novo after herbivore attacks but also after stimulation by abiotic stressors. They have some metabolic costs but they make the plant phenotypically plastic and herbivore adaptation more unlikely to occur [5].

HIPVs comprise isoprenoids, products of the lipoxygenase (LOX) pathway, such as GLVs, some carotenoid derivatives, indoles and phenolics, and phytohormones such as ethylene, jasmonic acid, and others [17,18]. It is hypothesized that forest VOCs are not only involved in some plant-related physiological functions (biogenic stress response, adaptation to climate changes), but they also have a central role in forest ecosystems (interplant communication, antimicrobial and insecticidal activity against parasites, influence on animals’ feeding behaviors as well as on underwood environmental conditions) [19]. Thus, forest VOCs can have several functions for plant physiology [18]; the most important of them are summarized in Table 1 and Figure 1.

### 1.2. Non-Tree-Derived Forest BVOCs

In temperate forests, the canopy layer of trees acts as the largest emission source of BVOCs [19] and, in general, from a quantitative point of view, plant leaves with their canals, oil glands, and glandular trichomes are the major emitters of these volatile compounds [8,10]. However, all plant organs and tissues produce BVOCs: leaves, flowers, and fruits release them into the atmosphere, whereas roots secrete them into the soil. Moreover, wood, phloem, and bark of trunks can serve as pools of stored BVOCs [19].

Nevertheless, BVOCs are not exclusively tree-derived: soil bacteria, mycorrhizal fungi, and other rhizosphere microbes can also emit BVOCs, but it is not easy to evaluate the exact contribution of these non-plant sources of BVOCs in the soil or their role in the rhizosphere. This is because their emissions cannot be easily separated from those of the roots [19] and because of experimental difficulties [8]. These limits notwithstanding, one study showed that the emission of sesquiterpenes (in particular, (–)-thujopsene) by fungi (*Laccaria bicolor*) can interact with the ability of *Populus* trees to develop their root system (in particular, with the proliferation of their lateral roots) [20]. One research article reported that, in the Amazonian rainforest, soil microorganisms can emit large amounts of sesquiterpenes, and these emissions strongly influence atmospheric chemistry near the soil surface and under the canopy, to an extent comparable with canopy emissions themselves [21]. Šimpraga and colleagues reported that there is no similar assessment for temperate and boreal forests, but there are limited data on bacterial VOC profiles (rich in alkenes, alcohols, ketones, and terpenes) and fungal VOC profiles (dominated by alcohols, benzenoids, aldehydes, and ketones) [19]. Apart from soil-derived emission, other, quantitatively less important, BVOC emissions in the forest can originate from understorey vegetation. In an article relative to subarctic forests of *Betula pubescens* ssp. *czerepanovii* (mountain birch), the authors found that BVOCs emitted by the understorey vegetation of *Rhododendron tomentosum* are adsorbed by tree stands growing above them, thus affecting the volatile emission profile of mountain birches [22].

Overall, as suggested by these limited data, trees are considered as the major emitters of BVOCs in temperate forests and the contribution of non-tree emitters is still difficult to properly characterize in sufficient detail, thus probably having a more indirect effect on the composition of the forest air. For these reasons, it was decided to focus this review on tree-derived forest VOCs.

### 1.3. Research Aim

The primary aim of this research work is to outline evidence-based physiological effects of forest VOCs in order to evaluate their potential integrated action on health. A secondary aim is to propose strategies to effectively and sustainably harness any of their beneficial effects in real-life contexts. Among all forest VOCs, it has been decided to focus our attention on pinene and limonene due to their wide uses and applications, as well as to the fact that their effects on health have been better studied [23,24]. The chemistry and diversity of forest VOCs (volatile organic compounds) is also analyzed for better comprehensiveness.

## 2. Materials and Methods

This research work was designed as an overview of the scientific literature with a critical discussion of retrieved evidence [25]. General recommendations to improve the quality of narrative reviews were taken into account for better structuring this research [26,27]. Scientific databases like Medline, PsycINFO, Cochrane Library, and Google Scholar were searched for articles on the topic with relevant keywords, including “volatile organic compound*”, “forest air”, “forest atmosphere”, “forest bathing”, “phytoncide*”, “pinene”, “limonene”, “psycho*”, “neuro*”, “endocrine*”, “hormone*” “immune*”, “prevention”, “inflammation”, “stress”, and “health”. The search yielded 4159 results (1211 from Medline, 200 from PsycINFO, 158 from Cochrane Library, and 2590 from Google Scholar) and, after article screening, 147 studies were considered eligible to be included and discussed in this review. Considering that evidence on the topic is limited, cross-disciplinary, and highly heterogeneous, it was decided to structure the article as a narrative qualitative synthesis of the scientific literature in order to provide the reader with a state-of-the art dynamic description of the topic, stemming from a combination of retrieved evidence with the specific professional background and expertise of each author. Beyond the basis of available evidence, it was decided to collect and discuss ideas, hypotheses, current examples, and future perspectives for research and practice. In particular, the focus on health effects of forest VOCs (especially limonene and pinene) led to highlighting their potential activity on stress and inflammation. Finally, three levels of discussion were considered, with a wide range of implications varying from clinical activities to policymaking: individual wellbeing, public health, and landscape design.

## 3. Results

### 3.1. Biochemistry of Forest VOCs

Forest VOCs are the product of many different metabolic pathways: the isoprenoids are the most important group of metabolites in terms of diversity of compounds and quantities of emissions, but the products of the LOX pathway as well as shikimate derivatives are also highly relevant. Main biosynthetic pathways for BVOCs are graphically displayed in Figure 2.

#### 3.1.1. Isoprenoids

The volatile isoprenoids include isoprene (C5), monoterpenes (C10), homoterpenes (C11 or C16), and sesquiterpenes (C15), and they all derive from the precursor dimethylallyl diphosphate (DMAPP) and its isomer isopentenyl diphosphate (IPP). These precursors are synthesized by two separate pathways, one active in plastids (the deoxyxylulose-5-phosphate (DXP), also known as the methylerythritol phosphate (MEP) pathway), and the second in the cytosol of the secretory cells themselves (the mevalonate (MVA) pathway) [17]. In fact, IPP and DMAPP seem to originate in the plastids from the MEP pathway, while leucoplasts contain the enzymes for the first steps of monoterpene biosynthesis. In the MVA pathway, DMAPP accepts one IPP (by prenyltransferase—PT), leading to the production of farnesyl diphosphate (FPP) and, consequently, to sesquiterpenoids (although, in some instances, there is a very small production of monoterpenes) and to the homoterpene DMTT. In the MEP pathway, the hemiterpene isoprene is synthesized from DMAPP alone by terpene synthase or cyclase (TPS), and the fusion of DMAPP to one or more IPPs leads to the production of geranyl diphosphate (GPP) (precursor of the monoterpenoids) and geranyl geranyl diphosphate (GGPP), which leads to the synthesis of diterpenes from which the homoterpene TMTT derives. Although the MEP pathway is plastid-located, the enzymes necessary for sesquiterpene synthesis are cytosolic. Hence, MEP-derived IPP or DMAPP need to travel from plastids to the cytosol [17,18]. The biosynthesis of the homoterpenes DMNT and TMTT follows a less direct pathway. DMNT derives from the oxidative degradation of (E)-nerolidol (a C15-alcohol), and TMTT from the oxidative degradation of (E,E)-geranyl linalool (a C20-alcohol) [28].

#### 3.1.2. Oxylipins

Oxylipins are products of the lipoxygenase (LOX) pathway. They derive from the polyunsaturated C18 fatty acids of the chloroplast membrane (such as 13-hydroperoxylinolenic acid), which, in times of stress or damage, are cleaved by 9-LOX and 13-LOX pathways to give highly reactive products like 6-carbon aldehydes and alcohols (GLVs) and derivatives of jasmonic acid such as methyl jasmonate [9].

#### 3.1.3. Shikimate Pathway

Another important group of BVOCs consists of compounds containing an aromatic ring, such as indole (derived from reactions of modified amino acids or their precursors) or the derivatives of phenylalanine, either via the phenylpropanoid pathway (eugenol and methyl chavicol) or the benzoate pathway (methyl salicylate) [17] (Figure 2).

### 3.2. A Qualitative and Quantitative Analysis of Forest VOC Emissions

The composition of forest VOCs (probably the key VOC source [11]) is expected to consist mainly of constitutive compounds, but, since it cannot be excluded that, at any given moment, some individual plants in the forest may be under herbivore attack or abiotic stress, and provided that the rate of VOC emissions can be many times higher than baseline after a herbivore attack [29], a part of the total forest emissions can consist of biotic and abiotic-induced VOCs. This can be important for the evaluation of biological activities of forest VOCs because the range of biotic-induced compounds is often highly different from that of constitutive ones and can be also different from the abiotic-induced ones [29,30]. If seen from the plants’ functional point of view, VOCs thus comprise all volatile constitutive defense compounds (volatile phytoanticipins, or phytoncides, as defined by Tokin), plus volatile inducible defense compounds (volatile phytoalexins or HIPVs) and non-defense compounds (attractants of pollinator and of seed dispersals).

The biochemical composition of forest air tends to follow diurnal variations, with BVOCs concentration usually peaking early in the afternoon and, especially in summertime, even in the morning (around 2 h after dawn), and it can be markedly influenced by several factors, including meteorological conditions, altitude, seasons, sunlight exposure, and tree types [31,32]. For example, it has been demonstrated that the air of forests dominated by conifer trees has a higher concentration of VOCs if compared to that of forests with more deciduous trees [31]. However, beyond these differences, it is estimated that, from a biochemical point of view, the majority of forest VOCs are isoprene and terpenes, especially monoterpenes, including limonene or pinene [33]. Specifically, according to Guenther and colleagues [3], isoprene dominates the emissions, with around 50% of the total BVOC emissions of 1 billion tons C/year. C1 and C2 oxygenated compounds such as methanol and acetaldehyde and C10 hydrocarbons such as α-pinene, β-pinene, (E)-β-ocimene, and d-limonene contribute to another 30%, while the remaining 20% is dominated by 20 compounds (mostly terpenoids), with the other 100 compounds accounting for a mere 3%.

Evidence from research done on open-field emissions of both conifer and deciduous forests [34,35,36], and on un-detached leaves or branchlets [30,37], supports these estimates, indicating that, apart from isoprene, the most common and dominant forest VOCs in terms of percentage of the total emissions are monoterpenes such as α-pinene, d-limonene, and β-pinene, followed by sabinene, myrcene, and camphene. Monoterpenes’ global annual emissions amount to 330–480 million tons, with an average atmospheric concentration above forests from some pmol/mol to several nmol/mol. Other quantitatively important forests VOCs such as GLVs and acetaldehyde are emitted in quantities of around 260 million tons and show atmospheric concentrations of around 1–3 nmol/mol [11,16].

From a quantitative point of view, studies which aimed to determine the atmospheric concentration of some categories of BVOCs under forest canopy reported heterogeneous findings, probably due to a significant influence of the abovementioned factors on the forest air composition: approximately, on the basis of available evidence, the total amount of terpenes can vary from a minimum of 3.5 to a maximum of 56.0 μg/m^3^, isoprene can range from 0.1 to 28.0 μg/m^3^, and monoterpenes may have an average concentration of 4.5 μg/m^3^, whereas the amount of LOX pathway products is usually below the upper threshold of 3.4 μg/m^3^ [16,31,32,34,38,39,40,41,42,43,44] (Figure 3). In particular, if we only consider monoterpenes, the concentration range of seven of them under the canopy of a Scots pine forest was reported in a study [44] (Figure 4). For example, limonene concentration can vary from 0.81 to 8.10 μg/m^3^, while α-pinene can reach higher concentrations in pine forest air, up to 13.05 μg/m^3^ [44]. However, as reported in a review endorsed by the World Health Organization (WHO), measured concentrations of limonene in the air of different forest areas across the world (Europe, North America, and Asia) can largely vary and even reach values of 12.2 g/m^3^ [45]. Therefore, it is important to consider this high degree of variability when studying health and environmental effects of forest VOCs.

Studies on emissions from plants after artificial lesions [46,47], or experiments performed using techniques such as branch enclosures, which are likely to produce stress in the plant [48], show the presence of GLVs like cis-3-hexen-1-ol, cis-3-hexenyl acetate, cis-3-hexenyl butyrate, and cis-3-hexenyl valerate, although there are contradictory examples [49]. This fits well with the fact that C6 volatiles (aldehydes, alcohols, and their derivatives) such as GLVs are derived from cell membrane denaturation [12]. Not only direct physical stress but also temperature variations can affect the release of BVOCs. GLVs, as well as C1 and C2 oxygenated compound emissions, can increase during heat stress and might continue even for a long time after the temperature has returned to normal levels. Trees with specific storage structures (such as resin ducts in most of the conifers) are less affected by changes in temperature and are moderate isoprenoid emitters, unless these structures are damaged by insects, fires, strong wind, or high humidity. Most deciduous species do not have such structures and release BVOCs directly from mesophyll cells in a more light- and temperature-dependent manner (in particular, *Fagaceae* and *Salicaceae*) [12]. However, the most typical compounds emitted after biotic stress are the homoterpenes TMTT and DMNT [17].

Overall, the diversity of BVOCs is certainly much higher, and this is represented in Table 2, in descending order of quantity of emissions [4,16,30,34,35,37,42,46,47,48,49,50,51].

### 3.3. BVOCs and Plant-Derived Essential Oils

Apart from being plant-emitted components of the atmosphere, forest VOCs can be found in plant-derived products and, in particular, in distilled essential oils, the lipophilic result of steam distillation of aromatic plants [52]. Although essential oils do contain BVOCs, not all BVOCs are present in essential oils, and some molecules included in essential oils are not part of the BVOC molecular suite but are rather artefacts of distillation. Essential oils are dominated by volatile terpenes and terpenoids (C10 and C15) and, in specific cases, they contain phenylpropanoids, oxylipins, and, more rarely, N- and S-containing molecules. They do not contain appreciable quantities of isoprene, highly reactive chemical species such as GLVs (although occasionally found in essential oils from *Camellia sinensis*, *Viola odorata*, *Carpinus betulus*, *Fragaria vesca*, *Rubus idaeus*), or any of the volatile but hydrophilic compounds found in hydrosols [53]. Essential oils have a higher percentage of heavier compounds (sesquiterpenes and sesquiterpenoids) compared to atmospheric BVOCs, due to higher temperatures involved in distillation that allow vaporization of the less volatile compounds (the high-boiling ones), and are mainly obtained from plant species with storage structures (glandular trichomes, resiniferous ducts, etc.). Many essential oils seem capable of exerting interesting pharmacological activities, even if not all essential oils are active and no single essential oil exerts all these activities. Moreover, given that essential oils are inherently perceptively salient, they can have mood-altering effects, which are mediated not by classical pharmacological mechanisms but by olfactory images.

A simplified list of the most common and noteworthy activities of essential oils upon inhalation includes general antimicrobial ones and specific effects [54,55,56,57]:antimicrobial effects on antibiotic-resistant bacterial strains;antitussive, mucoactive, bronchodilation, and antispasmodic activities on the respiratory system;non-olfactory-mediated psychopharmacological effects on arousal, activation, memory loss, dementia, cognitive performance, anxiety, quality of life, quality of sleep;antioxidant effect;antinociceptive, anti-inflammatory, and cytotoxic activity;anti-nausea and spasmolytic effects on the intestine.

### 3.4. Evidence about the General Effects of Forest VOCs on Health

From a pharmacokinetic point of view, when visiting a forest, monoterpene VOCs such as limonene and pinene are mainly absorbed through inhalation, their blood levels rapidly rise after exposure, and they are mostly eliminated unchanged both in exhaled air and in the urine [58]. Accumulation in the adipose tissue has been demonstrated for some BVOCs like limonene [59], as well as some degree of metabolization for terpenes like pinene [60]. Their general activity on inflammation, stress, sleep, and psychological behavior is described below. In particular, among monoterpenes, limonene and pinene are important for their biological activities even on human subjects [36,58] and, therefore, their health properties have been described in dedicated paragraphs.

#### 3.4.1. Immune System and Inflammation

A recent review synthesized available evidence on the immune function enhancing effects of essential oils and, in particular, results of studies about forest bathing were discussed, thus suggesting that even 2-h-long forest walks can be associated with a significant increase in Natural Killer (NK) cell count and activity and can augment the number of cytolytic molecules expressing cells, with such variations lasting for many days after the experience, both in healthy subjects and in individuals with cancer-related pain [61]. Some studies included in this review also showed an effect of forest bathing on serum levels of IL-8, IL-6, and TNFα [61]. These findings confirmed the main results of preclinical laboratory experiments [62]. As with all studies about forest bathing, the suggestion that the mechanism for the observed health benefits is due to the inhalation of forest BVOCs must be balanced by the possible role of mediators like other sensory or non-sensory stimulators [63].

A review of preclinical studies analyzed available evidence regarding the effects of the most common volatile terpenoids found among BVOCs on inflammation and concluded that these compounds might act as inhibitors of pro-inflammatory mediators such as NO, TNF-α, and PGE2, thus modulating signal transduction pathways involving transcription factors like NF-κB and the mitogen-activated protein kinase (MAPK) [4]. Moreover, some terpenoids can directly interact with TRP receptors (implicated in nociception and inflammatory responses), reduce oxidative stress, and stimulate cell autophagy. The authors reported that, although these activities are not limited to forest bathing and can be exploited through skin applications or oral intake of essential oils, the contact during forest bathing may be safer, although less beneficial [4]. In any case, only a small number of preclinical studies have investigated the role of total extracts of forest BVOCs: volatile compounds extracted from hinoki wood essential oil have been reported to increase in vitro the natural killer (NK) cell activity by enhancing the expression of cytolytic enzymes [64]; also, an extract obtained from pinecones has been found to protect bovine mammary epithelial cells, stimulated with lipopolysaccharide (LPS), from oxidative stress after downregulating cyclooxygenase-2 (COX-2) expression [65]. On the other hand, single terpenes (α-phellandrene, 1,8 cineole, linalool, and β-caryophyllene) have been successfully used as pre-treatment to reduce the levels of interleukins, TNF-α, and iNOS in cellular and animal inflammatory-induced models [66,67,68,69]. Furthermore, mitogen-activated protein kinase (MAPK) signal transduction pathways (e.g., JNK and p38) can be inhibited by myrcene, limonene, and borneol [70,71]. Interestingly, β-caryophyllene has been shown to bind the peripheral receptor for cannabinoids (CB2R) and modulate the immune function [72,73]. Additionally, terpene treatment is able to reduce nucleus translocation, phosphorylation, and expression levels of NF-kB [4] and Nrf2 [74,75,76], the main transcriptional factors regulating levels of pro-inflammatory mediators. Moreover, many studies have demonstrated some BVOCs’ (i.e., myrcene, linalool, 1,8 cineole, α- and γ-terpinene) antioxidant properties such as the decrease of Reactive Oxygen Species (ROS), Matrix Metallo-Proteinase (MMP), and Nitric Oxide (NO) production [77,78,79,80]. Finally, inhibition of neuro-inflammation has been shown in vitro for 1,8-cineole [81] and β-caryophyllene [82,83] and in in vivo animal models for d-limonene [84] and linalool [85].

#### 3.4.2. Nervous System and Psychological Behavior

A systematic review on the general effects of forest bathing found that this meditative practice can be associated with positive health benefits, like a reduction in heart rate and blood pressure, increased relaxation, ameliorated general wellbeing, and improvement of depression scores [86], although available evidence is still limited and observed effects are likely to be multifactorial, thus probably depending not only on the presence of BVOCs in the atmosphere but also on the whole natural, green context [87]. These results are in line with preclinical studies on animal behavioral models investigating BVOCs’ neurological role in central nervous system (CNS) depression. In fact, BVOCs have been able to reduce locomotor activity and increase muscle relaxation, with consequent improvement of sleep, pain, and anxiety in mice [88,89,90]. Recently, it has been reported that substances derived from pine trees, such as α-pinene and 3-carene, can enhance sleep by acting as a positive modulator for GABA-A-BZD receptors [91]. Similar effects are reported for borneol, verbenol, isopulegol, and pinocarveol [92,93]. Moreover, inhaled linalool and β-pinene have shown anxiolytic [94,95] and antidepressive [96,97] properties in different animal models.

#### 3.4.3. Endocrine System and Stress

A systematic review concluded that forest bathing can induce a state of physiological relaxation, with a concomitant reduction in stress hormone levels, and it can also exert beneficial effects on general stress-related health outcomes, such as short-term improvements of cardiovascular and metabolic parameters, including reduced glucose levels [98]. A very recent systematic review of 22 clinical studies analyzing the effects of forest bathing on stress discovered that salivary cortisol levels were significantly lower in forest groups compared to control groups both before and after the intervention and pointed out that, although further research is needed, forest bathing can significantly influence cortisol levels in the short term in such a way as to reduce stress, with anticipated placebo effects playing an important role in this [99]. In animal models, forest VOCs have shown a positive effect on stress response [100,101], whilst levels of stress hormones (i.e., plasma corticosterone and adrenaline) have not been influenced by limonene and α-pinene [102,103]. On the other hand, limonene may protect PC12 cells against corticosterone-induced neurotoxicity via the modulation of AMP-activated protein kinase cascade [104].

### 3.5. Limonene

Limonene (1-methyl-4-prop-1-en-2-ylcyclohexene) is a monocyclic monoterpene hydrocarbon, a cyclohex-1-ene substituted by a methyl group at position 1 and a prop-1-en-2-yl group at position 4, respectively. Given that limonene has a carbon chiral center at position 4, it occurs as two optical isomers: D-limonene or [R-(+)-isomer] and L-limonene or [S-(+)-isomer]. Of the two, D-limonene (also called “citrene” or “carvene”) is the most abundant in plants and it is typical of *Rutaceae* such as orange, lemon, mandarin, etc. On the other hand, L-limonene is far less common and it can be found in essential oils obtained from *Pinus* spp., *Illicium verum*, and *Mentha* spp., while the D-L mixture (dipentene) is even less common [53,105]. Table 3 reports average contents in D-limonene of some essential oils derived from trees [106].

#### 3.5.1. Anti-Inflammatory Activity

In a systematic review investigating the anti-inflammatory potential of forest BVOCs, D-limonene was shown to be able to modulate many pro-inflammatory mediators such as TNF-α, NO, IL-1β, IL-6, enzymes involved in the inflammatory response (5-LOX, COX-2, iNOS), transcription factors (NF-κB), and signal transduction kinases (p38, JNK, ERK) [4]. In particular, the modulation of inflammatory substances and transcription factors was demonstrated in various animal models (Wistar rats, BALB/c mice, fruit flies) and in some cell cultures (raw 264.7 cell lines, human chondrocytes) [4]. These activities seemed responsible both for preventative action and for a direct effect on inflammation [24]. It was also noted that the (*l*)-(+)-limonene enantiomer was approximately three-fold less active than the (*S*)-(–)-limonene one in 5-lipoxygenase inhibitory activity [107].

On the specific role of limonene in respiratory tract inflammation, the review by Santana and colleagues concluded that the monoterpene has effective anti-inflammatory activity in both preventing and controlling respiratory system injuries [108], and this points towards a possible role for limonene in the management of allergic airway diseases and asthma [24].

In a clinical trial on “inflamm-ageing” and oxidative stress in healthy elderly subjects, the intake of 10 mg/kg of limonene as a food supplement in the context of a specific diet resulted in a decrease in fibrinogen, glucose, and insulin levels and in an amelioration of the HOMA-IR index, a homeostatic model assessment used to quantify insulin resistance and beta-cell function [109].

#### 3.5.2. Antioxidant Activity

In a review conducted by Vieira and colleagues, limonene was described as an antioxidant compound with free radical scavenging properties, capable of reducing oxidative stress in diabetic rats and of attenuating oxidative stress not only in murine lymphocytes but also in human epithelial cells and fibroblasts, with cell proliferating, anti-apoptotic, and DNA-protecting effects [24]. Specifically, limonene may act by inhibiting caspase-3/caspase-9 activation, p38 MAPK phosphorylation, and LOX activity and by increasing the activities of cell antioxidant enzymes and the Bcl-2/Bax ratio [4].

#### 3.5.3. Antiproliferative Activity

D-Limonene has been studied as an antiproliferative agent, and in vitro experiments have suggested a series of possible mechanisms, such as the induction of phase II carcinogen-metabolizing enzymes (a); the reduction of cancer cell proliferation via prenyl-transferase inhibiting activity (possibly diminishing the addition of FPP to the tail of RAS proteins) (b); increased tumor cell apoptosis, autophagy (via the mitochondrial death pathway, the PI3k/Akt pathway, and the caspase-3 and -9 activity), and differentiation (c); growth suppression (through a reduction in cyclin-D1 and an increase in TGF-β signaling leading to a more benign phenotype) (d); decreased tumor-induced immunosuppression, tumor-promoting inflammation, angiogenesis, and metastasis (through a reduction in circulating VEGF and a blockage of the receptor VEGFR1) (e); improved genome stability (increased DNA damage repair and PARP cleavage) (f) [4,17,24,110,111]. In spite of the many mechanistic studies, there is a dearth of preclinical studies with proper animal models and of human clinical trials. The most relevant ones are summarized in Table 4 and mostly involve the oral administration of limonene.

#### 3.5.4. Antinociceptive Activity

Data about the activity of D-limonene on nociception are scant. Available evidence with laboratory animals points towards a bimodal action of this monoterpene, which, when administered orally, appears to inhibit nociception and to prevent artificially-induced hyperalgesia (probably by reducing the synthesis of cytokines), while, when topically applied, it seems capable of eliciting pain, probably after an interaction with TRPA1 ion channels [24,116].

#### 3.5.5. Other Pharmacological Activities

According to a review by Vieira and colleagues analyzing data from experiments with animal models, limonene may be a potential metabolic agent, capable of improving hyperglycemia, pancreatic β-cell mass, lipid blood levels, and liver fat accumulation [24]. Limonene could also be a solvent for cholesterol, and it has been used to dissolve gallstones after direct infusion into the gallbladder [117]. Limonene may even exert protective and healing effects in experimental models of gastric ulcer and colitis, possibly by promoting mucus secretion and PGE2 production [24]. In in vitro settings with Rolf B1.T cells, limonene can decrease levels of neuropeptide Y, the most important appetite regulator [118]. It finally shows moderate antifungal and antibacterial activities, more pronounced in Gram-positive strains [116]. A summary of preclinical evidence of limonene biological activity has been reported in Table 5.

### 3.6. Pinenes

Pinene is a bicyclic monoterpene hydrocarbon with two structural isoforms, *α*- and *β*-pinenes. Both L- and/or D-forms, as well as racemic forms, may occur. These two isoforms are important constituents of conifer oleoresins and are widely distributed in plants, although *α*-pinene is generally more abundant.

α-pinene (2,6,6-trimethylbicyclo[3.1.1]hept-2-ene) is a bicyclo[3.1.1]hept-2-ene substituted by methyl groups at positions 2, 6, and 6, respectively. Both optical forms are present in conifer essential oils [138]. Table 6 reports the most common tree species whose total essential oil contains a quantity of α-pinene at a percentage of more than 20% [70].

β-pinene (6,6-dimethyl-2-methylidenebicyclo[3.1.1]heptane) is a bicyclo[3.1.1]heptane substituted by methyl groups at positions 6 and 6, respectively, and an exocyclic double bond [139]. It usually accompanies α-pinene in turpentine and other essential oils and is present with both optical forms. It is the main compound of essential oil derived from *Piper capense* [53]. Table 7 reports the most common tree species whose total essential oil contains a quantity of β-pinene at a percentage of more than 20% [70].

#### Pharmacological and Clinical Activities of Pinenes

In a recent review of the scientific literature, it has been reported that, although many in vitro studies exist about the pharmacological activities of pinene and pinene-containing essential oils, few in vivo studies and even less clinical trials on the topic have been conducted so far [23]. In fact, none of the reviewed clinical studies tested isolated pinene, but mixtures of different molecules or whole essential oils have been often tested.

A clinical trial involving patients with bronchitis tested a product containing α-pinene, D-limonene, and 1,8-cineole, and, although study results were positive [140], if we consider the clinically recognized effects of 1,8-cineole on bronchitis, it is impossible to discern whether α-pinene was important for the results.

A single-blind clinical trial on pediculosis tested two essential oils, one of *Eucalyptus globulus* and one of *Cinnamomum zeylanicum*, both containing α-pinene, and found eucalyptus oil to be superior to permethrin [141], while a second study evaluated the effects of essential oil obtained from the leaves of *Lippia multiflora* and found it to be superior to benzyl benzoate [142]. In both cases, however, given the recognized significant activity of 1,8-cineole and terpineol, respectively, on pediculosis, it is impossible to declare pinene as an active and useful compound per se.

The reviewed trials on memory and Alzheimer’s disease tested only whole essential oils belonging to the genus *Salvia* (*Salvia officinalis* and *Salvia lavandulaefolia*) for their inhibitory activity on cholinesterases [143,144,145]. Again, as previously noted, these essential oils contain, apart from pinene, several compounds, and, in fact, 1,8-cineole is likely to be the most active molecule, although a synergistic interaction with other monoterpenes has been suggested.

Amongst pharmacological effects of pinenes demonstrated on the basis of preclinical evidence, authors of the previously mentioned review specifically analyzed those relevant to cancer growth, allergies, inflammation, infections, and anxiety [23]. Data about cancer cells suggest that α-pinene might have effects on NF-κB and IκBα, and, therefore, on inflammation, mitochondrial functions, ROS production, caspase-3 activity, and DNA fragmentation [146], but also on efflux pumps responsible for multidrug-resistant tumors [147], as well as on cell cycle arrest via the cyclin-B protein [148]. A pre-treatment of allergic rhinitis models with α-pinene resulted in a significant reduction in clinical symptoms and inflammatory cytokine levels [149]. α-pinene also appears to exert moderate antimicrobial [150,151] and immune-modulating effects [61,62]. Finally, it has been shown that α-pinene binds to the GABA-A receptor, at the benzodiazepine binding site, thus prolonging GABAergic inhibitory signaling and suggesting an anxiolytic and sleep-enhancing effect [91]. A summary of preclinical evidence of α-pinene biological activity has been reported in Table 8.

## 4. Discussion

### 4.1. Implications for Individual Health

Inhaling forest VOCs can result in useful antioxidant and anti-inflammatory effects on the airways, and the pharmacological activity of some terpenes absorbed through inhalation may be also beneficial to promote brain function by decreasing mental fatigue, inducing relaxation, and improving cognitive performance and mood [164,165]. The effect of these volatile compounds found in forest air is not limited to an action on the respiratory system, but, after absorption through inhalation, they are supposed to exert systemic effects, such as those on the nervous system (relaxing, anxiolytic, and antidepressant action). Evidence from laboratory studies seems to confirm these effects, thus underscoring that, in particular, limonene and pinene can modulate the release of various cytokines (for example, TNF-α, IL-1, IL-6), inflammatory mediators (for example, NF-κB signal transduction pathway, MAPK, COX-2 activity), and neurotransmitters (for example, dopamine levels, action on GABA receptors) in such a way as to reduce inflammation and pain and to improve anxiety, mood, and sleep quality (Table 5, Table 8, and Table 9).

Although the magnitude of some of these pharmacological effects may be limited when inhaling such compounds during a short forest trip due to their relative atmospheric concentration (generally lower than that one prepared in experimental settings) and pharmacokinetics, even a mild action associated with psychophysical relaxation may be already beneficial for individual wellbeing. In fact, large studies aimed at thoroughly assessing pharmacokinetic and pharmacodynamic characteristics of forest VOCs in real-life situations are, to date, actually lacking, and it is only possible to formulate some plausible hypotheses. For example, with regard to limonene, human studies discussed in this review mostly investigated pharmacological effects of its oral intake when taken in relatively high quantities, up to several grams a day (Table 4). In a pharmacokinetic study coupled with a phase-I trial, the administration of limonene to cancer patients was observed to be followed within a few hours by a peak in limonene plasma concentrations of 10.8 μM (mean C_max_) when the oral daily dose amounted to 8 g/m^2^ and of 20.5 μM (mean C_max_) when the dose was 12 g/m^2^ [114]. However, inhaling limonene in a forest setting can be quite different, since many environmental and individual characteristics along with inhalation-related pharmacokinetic mechanisms can introduce a high degree of variability, which makes it hard (to date) to fully predict the exact short- and long-term health effects of a natural and real-life exposure to specific BVOCs. In fact, as reported in a review, measured concentrations of limonene in the air of different forest areas can largely vary, ranging from as little as 0.9 ng/m^3^ to as much as 12.2 g/m^3^ [45]. In a toxicokinetic study with healthy volunteers, participants were exposed for 2 h in a chamber with different concentrations of limonene (10, 225, and 450 mg/m^3^) and the pulmonary uptake was estimated to be up to 70% of the amount supplied, with plasma concentrations of this substance rapidly rising within 2 h after exposure and peaking at different levels (ranging, on average, from less than 2 to almost 25 μM), depending on the quantity of limonene diffused in the air [166]. No major short-term clinically symptomatic or toxic/irritative effects were observed, and only minor changes in respiratory function were reported when limonene concentration in the chamber was set at the highest experimental level (450 mg/m^3^) [166]. However, the pulmonary absorption of terpenes may be even less efficient, since more recent studies state that it could be 54–76% of the dose supplied with inhaled air [167]. In an interesting study involving a small cohort of healthy subjects, blood concentrations of some conifer-derived monoterpenes were measured before and after a 60-min trip in a Japanese forest [58]. The authors noticed that blood concentrations of some BVOCs exhibited a marked increase after visiting the forest, and, in particular, average plasma levels of α-pinene changed from 2.6 (baseline) to 19.4 nM (post-intervention) [58]. A pharmacokinetic study about metabolites of α-pinene showed that their blood levels tend to markedly and rapidly peak after the administration of a single oral dose of 10 mg to four volunteers, with trans-verbenol (one of the most relevant pinene metabolites) C_max_ quickly reaching a mean value of 26 nM and α-pinene being already undetectable 1 h after the exposure [60]. In brief, blood concentrations of BVOCs after indoor and, even more, outdoor inhalation show a higher degree of variability than after their oral intake in more controlled experimental settings. With available data, we can only affirm that plasma concentrations of BVOCs tend to rise whenever a subject is exposed to a forest, but, to date, due to limitations of existing studies, it is not possible to exactly quantify the actual magnitude of their specific beneficial effects for human health, especially with regard to their action at a systemic level.

Taking into account all these findings, potential moderators of the effect and confounding factors to be considered by researchers when planning future studies can include the following:the exact concentration of any BVOC in the forest air when exposure occurs;the duration of exposure;the intensity of physical activities performed in the green environment;specific modalities of biological sample collection, transport, and analysis;individual health-related characteristics, which can influence the uptake, metabolism, accumulation, and excretion of inhaled BVOCs;lifestyle habits, which can determine baseline blood levels of compounds like limonene and pinene absorbed from food, medicines, and perfumes [58].

Further rigorous pharmacokinetic studies ought to be performed to thoroughly estimate the actual absorbed quantity of BVOCs during a forest trip. Then, it would be possible to precisely assess short- and long-term effects on health of the inhalation of specific BVOCs and to correlate such effects with specific thresholds of BVOCs concentrations in the forest air. It is known that forest VOCs like limonene and, even more, pinene, are mostly released by conifers (pines, fir trees, cypresses) (Table 3, Table 6, and Table 7), and, therefore, the exposure to these volatile substances (with more documented effects on health) can be maximized by visiting a forest whose composition is rich in these trees. In general, beyond differences in tree composition, levels of BVOCs in the forest air exhibit diurnal quali-quantitative variations, with two daily peaks (early in the morning and in the afternoon) [31,32], and these cyclic changes may influence the magnitude and type of potential beneficial effects of their inhalation depending on the time of the day when the exposure occurs. For this reason, it would be useful to carry out additional environmental studies aimed at quantifying all concentration changes in forest VOCs in order to maximize the visitors’ exposure and, therefore, the impact of such compounds on health after inhalation.

Moreover, beneficial psychological and physiological effects of visiting a forest (such as in the meditative practice called “forest bathing”) cannot be solely attributed to VOC inhalation but are due to a global and integrated stimulation of the five senses [63], induced by all specific characteristics of the natural environment, with the visual component probably playing a fundamental role in the overall effect [99]. Therefore, even if inhaling BVOCs can exert useful effects on health while visiting a forest, absorbing these volatile compounds does not cover all benefits of forest bathing, which, to date, cannot be artificially replaced with a virtual experience and actually requires a full real-life interaction with the natural environment. However, further investigating all medicinal properties of specific forest VOCs with properly designed clinical studies can be useful to discover their potential uses in medical practice as drugs for some health conditions and to possibly discover new therapeutic agents derived from them.

**Table 9 ijerph-17-06506-t009:** Summary of preclinical evidence relative to biological activities of the five most common forest VOCs [3,30,34,35,36,37].

Molecule	Effects	Mechanisms	References
**D-Limonene**	Anti-inflammatory	It inhibits the synthesis or release of pro-inflammatory mediators (TNF-α, NO, IL-1β, IL-6, IL-5, IL-13, TGF-β), enzymes (5-LOX, COX-2, iNOS), transcription factors (NF-κB), and mitogen-activated protein (MAP) kinase family members (p38, JNK, ERK).	[4,70,71,124]
	Antioxidant	It inhibits caspase-3/caspase-9 activation, increases the activities of cell antioxidant enzymes (catalase, peroxidase) and the Bcl-2/Bax ratio.	[4,70,123]
	Antiproliferative	It induces phase II carcinogen-metabolizing enzymes, inhibits prenyl-transferase activity, increases cell autophagy (via MAP1LC3B, mitochondrial death pathway, the PI3k/Akt pathway, and caspase-3 and -9 activity) and differentiation, reduces cyclin-D1 and increases TGF-β signaling, decreases tumor-induced immunosuppression, reduces circulating Vascular Endothelial Growth Factor (VEGF) and blocks the receptor VEGF-R1, increases DNA damage repair and PARP cleavage. It modulates the expression of the chemotactic protein MCP-1 and of the proteolytic enzymes MMP-2, MMP-9.	[4,17,24,110,111,121,124,128,129]
	Antinociceptive	Bimodal activity: topically applied, it seems capable of eliciting pain, via interaction with TRPA1 ion channels, while in other modes of administration, it has shown antinociceptive effects in different experimental models.	[24,90,116,127,133,136,137]
	Anxiolytic	It shows some degree of efficacy in mice and rat models.	[90,127,135]
	Antidepressant	It shows some degree of efficacy in mice and rat models.	[132,133,134]
**α-Pinene**	Anxiolytic, sedative	In several animal models, it enhances sleep by acting as a positive modulator for GABA-A-BZD receptors, prolonging GABAergic inhibitory signaling.	[91,155,157,158,159,160,161]
	Anti-inflammatory	It modulates NF-κB and IκBα, ERK, JNK, IL-1β, IL-6, iNOS (and NO secretions), MMP-1, MMP-13, COX-2.	[146,152,153,155,156]
	Antioxidant	It reduces ROS production, caspase-3 activity, modulates superoxide dismutase, catalase, peroxidase activity, NO and IL-6 secretions.	[146,155,168]
	Antiproliferative	It acts on efflux pumps responsible for multidrug-resistant tumors and on cell cycle arrest via the cyclin-B protein.	[147,148]
	Analgesic	It shows some degree of efficacy in animal models.	[156,162,163]
**β-Pinene**	Anxiolytic, antidepressant	It binds to the GABA-A receptor, prolonging GABAergic inhibitory signaling. It showed efficacy in animal models.	[91,94,95,96,97]
	Anti-inflammatory	It modulates NF-κB and IκBα.	[146]
	Antioxidant	It reduces ROS production, caspase-3 activity, MMP and NO activities.	[77,78,79,80,146]
	Antiproliferative	It acts on efflux pumps responsible for multidrug-resistant tumors and on cell cycle arrest via the cyclin-B protein.	[147,148]
**β-Myrcene**	Anti-inflammatory	It modulates MAP kinases such as JNK, p38, and it inhibits the synthesis and release of PGE-2.	[70,71,169]
	Antiproliferative	It blocks hepatic carcinogenesis caused by aflatoxin.	[170]
	Analgesic	It is analgesic in mice, and its action is blocked by naloxone, perhaps via the α-2 adrenoreceptor.	[171]
	Sedative, myorelaxant	It is a muscle relaxant in mice, and it potentiates barbiturate sleep time at high doses.	[88]
	Gastroprotective	It contributes to the integrity of the gastric mucosa, decreasing ulcerative lesions, attenuating lipid peroxidative damage, and preventing depletion of GSH, GR, and GPx.	[172]
**Camphene**	Metabolism	As a food supplement, it reduces animal models’ body weight and increases adiponectin levels and receptor mRNA expression in the liver.	[173]
	Antiproliferative	It induces apoptosis in cancer cell lines (B16F10-Nex2 melanoma), chromatin condensation, cell shrinkage, apoptotic body formation, fragmentation of nucleus, and caspase-3 activation.	[174]
	Antioxidant	It prevents AAPH-induced lipoperoxidation and inhibits the superoxide radical.	[175]
	Antinociceptive	Weak effects on acetic acid-induced writhing in mice models.	[175]
	Antihyperlipidemic	It reduces total and LDL-cholesterol and triglycerides in hyperlipidemic rats and in HepG2 cells, not by inhibiting HMG-CoA reductase but by increasing apolipoprotein AI expression, possibly via SREBP-1 upregulation and MTP inhibition.	[176]

### 4.2. Implications for Preventive Medicine and Public Health

Useful effects of forests and green areas on psychophysical wellbeing have been underscored by several studies, with all sensory components, ranging from visual to olfactory stimulations, playing an integrated role in the overall beneficial action on individual health [63,86,99,177]. This may also explain the tight link between shamanism and wild environments, and why folkloristic and ritual practices have spread worldwide from the Amazon to the Siberian forest [178,179]. Evidence from research works analyzed in this review suggests that forest bathing can be beneficial for general health, thus exerting a general anti-stress and immune boosting activity. At a public health level, epidemiological studies have highlighted that there is evidence for a significant association between the extension of green areas within a given community and specific health outcomes like perceived mental and general health, as well as all-cause mortality [177]. Similar research works have confirmed a positive association between living closer to green areas and improvements in physician-assessed morbidity determined by various cardiovascular, muskuloskeletal, respiratory, mental, neurological, digestive, and other health conditions [180]. Indeed, the type of activities performed outside, possibly ranging from static relaxation to intense physical exercises, along with the subjective perception of safety and connectedness to nature while visiting the natural environment, can influence the magnitude of beneficial effects exerted by the exposure to green areas [181]. In particular, forests can be perceived as dangerous for several reasons (wild and unknown environment, potentially aggressive fauna, parasites, poisonous plants, limited supporting facilities, human activities like hunting, cutting trees, practicing extreme sports, etc.), and all necessary precautions have to be taken to prevent any risk of being harmed in order to fully and safely benefit from forest exposure. Additionally, when subjects suffer from specific diseases, a medical consultation is advised before planning to visit a forest.

Recently, research efforts of many expert scientists from different countries all around the world have converged into the definition of new concepts like “forest therapy” and “forest medicine”, which both refer to the scientific study and evidence-based applications of visiting a forest for therapeutic or preventive purposes [182]. In particular, from the foundation of the International Society of Nature and Forest Medicine (INFOM), researchers of this field have advocated for studying the topic in more depth, as well as to disseminate relevant findings worldwide in order to “establish Forest Therapy as an effective, evidence-based and low-cost public health treatment for selected lifestyle-related diseases” [182,183]. Beneficial effects of this practice are (although not exclusively) determined by the inhalation of forest VOCs, which are responsible for the olfactory perception of pleasing fragrances from trees and flowers [182]. Beyond these chemical factors, useful health effects of visiting a forest can be also attributed to physical (such as environmental characteristics, climate conditions, visual impressions, sounds) and psychological factors (to what extent the environment is perceived as pleasant, relaxing, and safe by an individual, i.e., placebo effects) [182]. Visiting forests on a relatively regular basis can be a good health-promoting practice, since, by reducing stress levels and boosting immune function, it seems capable of diminishing the incidence of stress-related and lifestyle-induced illnesses, varying from cardiovascular, respiratory, or metabolic diseases to neuropsychiatric conditions and, possibly, even cancer [86]. Some practical recommendations to optimally benefit from forest bathing for preventive purposes include performing only light physical activity in the natural environment (like walking at a regular pace without getting tired), stay well hydrated, avoiding the use of technological devices for recreation (thus preferring meditative contemplation as a relaxing practice during the forest visit and a bath in hot spring waters after the trip), spending in the forest 2 to 4 h and walking 2.5 to 5 km in a day along clean paths with some stopovers [182].

In general, on the basis of retrieved evidence and experts’ proposals, visiting a forest can be a useful practice, with the olfactory component playing a role in it, and it can be a sustainable public health practice to promote both individual wellbeing and community health, thus contributing to the prevention of many stress- and lifestyle-related diseases.

### 4.3. Forest VOCs and Natural Landscape Design

Today, around half of the world’s population lives in metropolitan areas and, over the years, technology, commuting, working, and some leisure activities have made us more and more inhabitants of an “indoor society”, in which natural light has been almost completely supplanted by artificial light, and the sounds and smells of nature are often a distant memory. Even if we do not notice it immediately, this can keep us under constant stress, not only because of the constant exposure to urban environments and their stressful stimuli but also because of the lack of contact with natural environments, which can promote health and wellbeing. Another equally important aspect is that green areas, even within an urban space, offer the possibility to organize open-air recreational activities, thus increasing socialization and residents’ quality of life [184,185,186].

In the last few decades, the concept of “biophilic design” has been introduced to indicate any sustainable design aimed at better connecting people with the natural world and with vital processes [187]. In particular, the attention of some outdoor designers has been focused on “healing gardens”, namely green areas purportedly planned to induce a psycho-physical state of wellbeing in their visitors/users [188]. Elements involved in the design of a healing garden are closely related to the therapy of the senses: perfumes (aromatherapy), colors (chromotherapy), sounds (music therapy), sometimes tactile or gustatory experiences, and symbols. Provided that, today, the relationship between man and architecture is mainly governed by sight, but, as underscored in this review, some natural volatile compounds can influence health, designing outdoor environments that have an impact on both sight and smell should be the next step. In particular, it is deemed necessary to create green areas and paths surrounded by fragrant shrubs, where it is possible to talk or relax, while walking immersed in nature with positive olfactory stimulations to guarantee an inclusive experience for all, both sighted and visually impaired or blind people. From this point of view, a precaution to take in order not to create confusion in passers-by is to group aromatic plants by species and arrange them at a certain distance, thus avoiding creating chaotic green spots. In fact, the relationship between space and smell is complex and depends on many factors, including the type of building materials, specific activities that take place in a given environment, the time of the day, air humidity, but also plants, animals, and people who pass through it. All these factors should be taken into account by landscape and outdoor designers, who must be encouraged to cooperate with a multidisciplinary team in order to achieve the best results.

A virtuous example of how to harness beneficial properties of forests in order to promote health is the landscape project involving the natural park of the Oasi Zegna, in Alta Val Sessera (Piedmont region, Italy). A study carried out in 2012 has scientifically shown that the beech wood of the Oasi Zegna can release volatile substances from the foliage, like monoterpenes, capable of bringing benefits to the organism and of stimulating the immune system in a positive way [189]. For this reason, three freely accessible paths were created from June to September, which corresponds to the period of maximum foliation of beech trees.

Another project designed by the CERFIT team (Research and Innovation Center in Phytotherapy and Integrated Medicine, Careggi University Hospital) is the so called “Giardino della Salute”, a healing garden which will be located in an outdoor green area (a pine forest) within the Careggi Hospital district in Florence (Italy) [190]. The ultimate aim of the project is to revitalize and redevelop a public park, as well as to improve the wellbeing and quality of life of all its users, with keen attention to healthcare personnel, hospital patients, and their relatives or visitors. In order to offer a personalized experience which is expected to meet individual needs, some specific footpaths will be laid: the “nutrition” path, where there will be plants used for food; the path of “the five senses”, characterized by an immersive sensory experience with various impressions; the “mind” path, for relaxing or stimulating the nervous system; the path of “natural hazards”, in which poisonous plants will be described for educational purposes; a path for generic users, another one dedicated to patients, and the last one to students. Specific characteristics of plants and flowers displayed along each path, including their appearance, colors, and phytochemical substances released into the surrounding environment, have been chosen to be synergically harnessed for beneficial purposes. The project also involves a herbalist and an educator who can guide visitors along their journey across the healing garden and help them to understand the most important uses and properties of medicinal plants. This project is a valuable example of how to make the most of beneficial properties of an ad hoc designed urban green area, taking into consideration various aspects of the effects of plants on human health, including the activity of biogenic VOCs released in the air.

## 5. Conclusions

Inhaling forest VOCs like limonene and pinene can result in useful antioxidant and anti-inflammatory effects on the airways, and the pharmacological activity of some terpenes absorbed through inhalation may be also beneficial to promote brain function by decreasing mental fatigue, inducing relaxation, and improving cognitive performance and mood. The tree composition can markedly influence the concentration of specific VOCs in the forest air. Moreover, beneficial psychological and physiological effects of visiting a forest (such as in the meditative practice called “forest bathing”) cannot be solely attributed to VOC inhalation but are due to a global and integrated stimulation of the five senses, induced by all specific characteristics of the natural environment, with the visual component probably playing a fundamental role in the overall effect. Globally, these findings can have useful implications for individual wellbeing, public health, and landscape design. Further clinical and environmental studies are advised, since the majority of the existing evidence is derived from laboratory findings.

## Figures and Tables

**Figure 1 ijerph-17-06506-f001:**
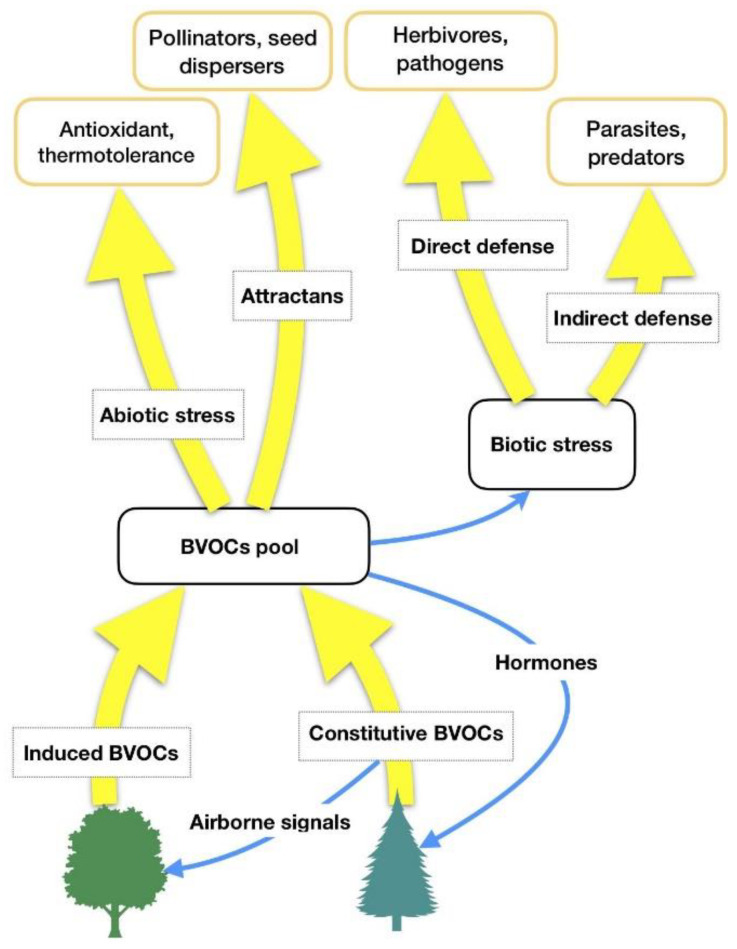
BVOCs’ functions in relation to biotic and abiotic stresses. Adapted from Laothawornkitkul et al. (2009) [8].

**Figure 2 ijerph-17-06506-f002:**
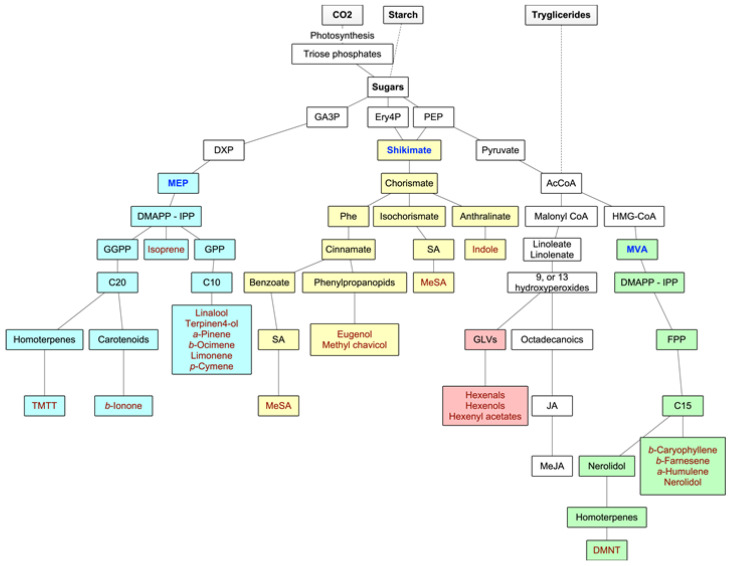
Main biosynthetic pathways for BVOCs. In green and yellow colors, the methylerythritol phosphate (MEP) and mevalonate (MVA) pathways to isoprenoids, respectively, in blue color, the lipoxygenase (LOX) pathway to GLVs, and in purple color, the shikimate pathway to aromatic compounds. Adapted from Maffei et al. (2011) [17].

**Figure 3 ijerph-17-06506-f003:**
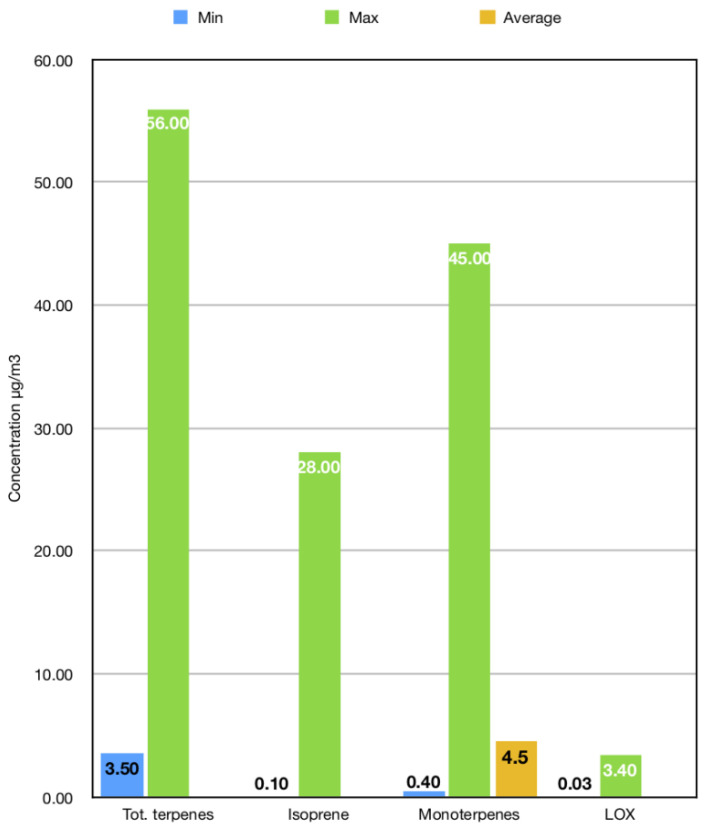
Atmospheric concentration of BVOCs under forest canopy [16,31,32,34,38,39,40,41,42,43,44]. Abbreviations: Tot. terpenes = total concentration of terpenes; LOX = products of the lipoxygenase (LOX) pathway.

**Figure 4 ijerph-17-06506-f004:**
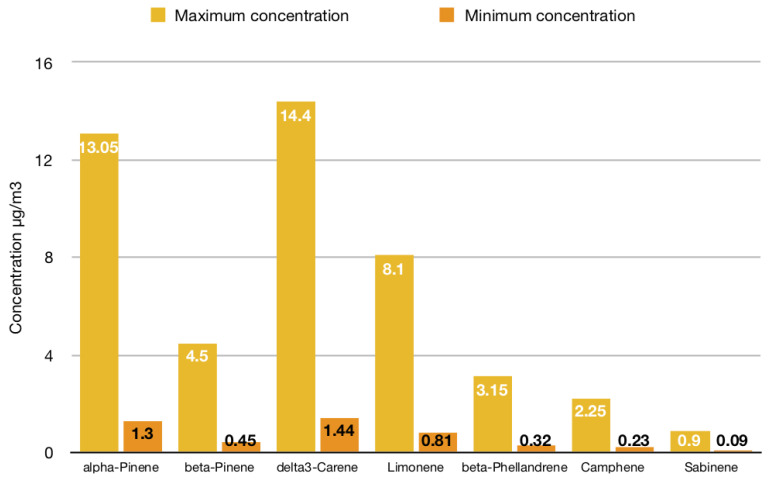
Atmospheric concentrations of seven monoterpenes under the canopy of a Scots pine forest [44].

**Table 1 ijerph-17-06506-t001:** Functions of constitutive and herbivore-induced forest Volatile Organic Compounds (VOCs).

Constitutive Forest VOCs	Herbivore-Induced Plant Volatiles (HIPVs)
Reduction of abiotic stress. Isoprene and monoterpenes increase general thermal tolerance of photosynthesis, protect photosynthetic apparatus and its activity under high-temperature stress by stabilizing the thylakoid membranes and quenching Reactive Oxygen Species (ROS)	Reduction of abiotic stress. Isoprene and monoterpenes increase general thermal tolerance of photosynthesis, protect photosynthetic apparatus and its activity under high-temperature stress by stabilizing the thylakoid membranes and quenching Reactive Oxygen Species (ROS)
Defense against herbivores. Comprises toxic, repellent, anti-nutritive constitutive BVOCs (biogenic Volatile Organic Compounds) or HIPVs, as well as growth and reproductive reducers	Defense against herbivores, mainly indirectly but also directly. HIPVs and volatile compounds that attract, nourish, or otherwise favor another organism that reduces herbivore pressure
Inter-plant signaling. HIPVs, especially Green Leaf Volatiles (GLVs), and constitutive BVOCs can travel from a herbivore-damaged part to other plants (both conspecific and heterospecific), activating defense genes and priming a more vigorous response after an attack	Inter- and intra-plant signaling. HIPVs, especially GLVs, and constitutive BVOCs can travel from a herbivore-damaged part to an undamaged one, or to other plants (both conspecific and heterospecific), activating defense genes and priming a more vigorous response after an attack
Defense against microbial pathogens	Defense against microbial pathogens
Allelopathy. Inhibition of competing species’ seed germination and competition	
Attraction of pollinators and seed dispersers	

**Table 2 ijerph-17-06506-t002:** Forest BVOCs and their chemical characteristics listed on the basis of the magnitude of emissions (in descending order).

Molecule	Chemical Family	IUPAC	Formula	Structure	CAS number	Boiling Point (at 760 mmHg)	Molar Mass (g/mol)	I/C ^1^	C/D ^2^	E ^3^	P ^4^
Isoprene	Isoprenoids	2-methylbuta-1,3-diene	C5H8	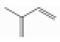	78-79-5	34.1 °C	68.12	C	D	****	+++
cis-3-Hexen-1-ol	GLVs	(*Z*)-hex-3-en-1-ol	C6H12O	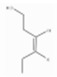	928-96-1	156.5 °C	100.16	I	D	***	+++
cis-3-Hexenal	GLVs	(Z)-hex-3-enal	C6H10O	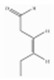	6789-80-6	126 °C	98.14	I	D	***	+++
cis-3-Hexenyl acetate	GLVs	[(Z)-hex-3-enyl] acetate	C8H14O2	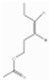	3681-71-8	174.2 °C	142.2	I	D	***	+++
d-Limonene	Monoterpene hydrocarbons	(4R)-1-methyl-4-prop-1-en-2-ylcyclohexene	C10H16	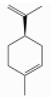	65996-98-7	175.4 °C	136.23	C, I	D	***	+/+++
α-Pinene	Monoterpene hydrocarbons	2,6,6-trimethylbicyclo[3.1.1]hept-2-ene	C10H16	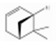	67762-73-6	156 °C	136.23	C, I	D	***	+/+++
(E)-β-Ocimene	Monoterpene hydrocarbons	(3E)-3,7-dimethylocta-1,3,6-triene	C10H16	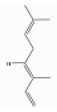	3779-61-1	175.2 °C	136.23	C, I	D	**	+/++
1,8-Cineole	Monoterpenoid ethers	1,3,3-trimethyl-2-oxabicyclo[2.2.2]octane	C10H18O	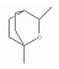	470-82-6	176.4 °C	154.25	C, I	D	**	
Camphor	Monoterpenoid ketones	1,7,7-trimethylbicyclo[2.2.1]heptan-2-one	C10H16O	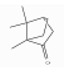	76-22-2	205.7 °C	152.23			**	
Linalool	Monoterpenoid alcohol	3,7-dimethylocta-1,6-dien-3-ol	C10H18O	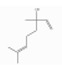	78-70-6	197.5 °C	154.25	C, L	D	**	+/++
p-Cymene	Aromatic monoterpene hydrocarbons	1-methyl-4-propan-2-ylbenzene	C10H14	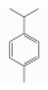	99-87-6	177 °C	134.22	C	D	**	
Sabinene	Monoterpene hydrocarbons	4-methylidene-1-propan-2-ylbicyclo[3.1.0]hexane	C10H16	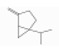	3387-41-5	164 °C	136.23	C	D	**	
β-Caryophyllene	Sesquiterpene hydrocarbons	(1R,4E,9S)-4,11,11-trimethyl-8-methylidenebicyclo[7.2.0]undec-4-ene	C15H24	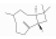	87-44-5	NA	204.35	C, I	D	**	+/++
β-Myrcene	Monoterpene hydrocarbons	7-methyl-3-methylideneocta-1,6-diene	C10H16	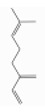	123-35-3	167 °C	136.23	C	D	**	
β-Pinene	Monoterpene hydrocarbons	6,6-dimethyl-2-methylidenebicyclo[3.1.1]heptane	C10H16	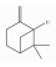	127-91-3	166.0 °C	136.234	C, I	D	**	
β 3-Carene	Monoterpene hydrocarbons	3,7,7-trimethylbicyclo[4.1.0]hept-3-ene	C10H16	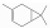	74806-04-5	171.4 °C	136.234	C	D	**	
(E)-Linalool-oxide	Monoterpenoid oxide	2-[(2S,5S)-5-ethenyl-5-methyloxolan-2-yl]propan-2-ol	C10H18O2	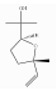	11063-78-8	NA	170.25	C	D	*	
(Z)-Linalool-oxide	Monoterpenoid oxide	2-(5-ethenyl-5-methyloxolan-2-yl)propan-2-ol	C10H18O2	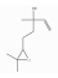	14049-11-7	224.2 °C	170.25	C	D	*	
Borneol	Monoterpenoid alcohol	1,7,7-trimethylbicyclo[2.2.1]heptan-2-ol	C10H18O	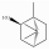	464-45-9	212.0 °C	154.25	C		*	
Bornyl acetate	Monoteropene-derived ester	(1,7,7-trimethyl-2-bicyclo[2.2.1]heptanyl) acetate	C12H20O2	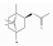	20347-65-3	223.5 °C	196.286	C		*	
Camphene	Monoterpene hydrocarbons	2,2-dimethyl-3-methylidenebicyclo[2.2.1]heptane	C10H16	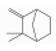	79-92-5	157.5 °C	136.234	C	D	*	
Terpinen-4-ol	Monoterpenoid alcohol	4-methyl-1-propan-2-ylcyclohex-3-en-1-ol	C10H18O	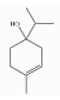	562-74-3	209.0 °C	154.25	C		*	
α-Copaene	Sesquiterpene hydrocarbons	(1R)-1,3-dimethyl-8-propan-2-yltricyclo[4.4.0.02,7]dec-3-ene	C15H24	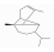	3856-25-5	248.5 °C	204.35	I		*	
α-Humulene	Sesquiterpene hydrocarbons	(1E,4E,8E)-2,6,6,9-tetramethylcycloundeca-1,4,8-triene	C15H24	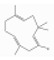	6753-98-6	166-168 °C	204.35	C	D	*	
α-Phellandrene	Monoterpene hydrocarbons	2-methyl-5-propan-2-ylcyclohexa-1,3-diene	C10H16	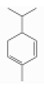	99-83-2	171.5 °C	136.234	C		*	
α-Terpinene	Monoterpene hydrocarbons	1-methyl-4-propan-2-ylcyclohexa-1,3-diene	C10H16	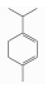	99-86-5	174.1 °C	136.234	C	D	*	
α-Terpineol	Monoterpenoid alcohol	2-(4-methylcyclohex-3-en-1-yl)propan-2-ol	C10H18O	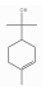	10482-56-1	217.5 °C	154.249	C		*	
α-Terpinolene	Monoterpene hydrocarbons	1-methyl-4-propan-2-ylidenecyclohexene	C10H16	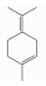	1124-27-2	186.0 °C	138.25	C	D	*	
β-Phellandrene	Monoterpene hydrocarbons	3-methylidene-6-propan-2-ylcyclohexene	C10H16	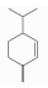	555-10-2	175 °C	136.234	C, I		*	+/+++
β-Terpinene	Monoterpene hydrocarbons	1-methyl-4-propan-2-ylcyclohexa-1,4-diene	C10H16	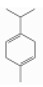	99-85-4	183.0 °C	136.234	C, I	D	*	
(Z)-β-Ocimene	Monoterpene hydrocarbons	(3Z)-3,7-dimethylocta-1,3,6-triene	C10H16	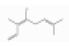	13877-91-3	175.2 °C	136.234	C, I	D		+/++
Bergamotene	Sesquiterpene hydrocarbons	6-methyl-2-methylidene-6-(4-methylpent-3-enyl)bicyclo[3.1.1]heptane	C15H24	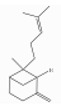	7663-66-3	NA	208.38	I			
DMNT	Homoterpene hydrocarbons	4,8-dimethylnona-1,3,7-triene	C11H18	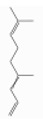	19945-61-0	195.6 °C	150.26	I			+/++
Longifolene	Sesquiterpene hydrocarbons	3,3,7-trimethyl-8-methylidenetricyclo[5.4.0.02,9]undecane	C15H24	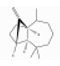	475-20-7	252.2 °C	204.35	C	D		
Methyl jasmonate	Jasmonate ester	methyl 2-[(1R,2R)-3-oxo-2-[(Z)-pent-2-enyl]cyclopentyl]acetate	C13H20O3	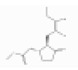	39924-52-2	302.9 °C	224.296	I			
Methyl salicylate	Benzoate ester	methyl 2-hydroxybenzoate	C8H8O3	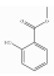	119-36-8	222.0 °C	152.147	I			++++
TMTT	Homoterpene hydrocarbons	(3*E*,7*Z*)-4,8,12-trimethyltrideca-1,3,7,11-tetraene	C16H26	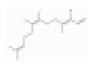	62235-06-7	293.2 °C	218.38	I			+/++
α-Thujene	Monoterpene hydrocarbons	2-methyl-5-propan-2-ylbicyclo[3.1.0]hex-2-en	C10H16	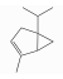	2867-05-2	152 °C	136.23	C			
β-Farnesene	Sesquiterpene hydrocarbons	7,11-dimethyl-3-methylidenedodeca-1,6,10-triene	C15H24	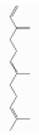	18794-84-8	279.6 °C	204.35	I			++

^1^ I/C = Inducible/constitutive: forest VOC types. ^2^ C/D = Conifers/deciduous: tree types. ^3^ E = Emissions **[3,8,42]**: highly abundant emissions: ****; abundant emissions: ***; moderately abundant emissions: **; common emissions: *. ^4^ P = Persistence **[3,8,42]**: less than 10 min: +; 10–60 min: ++; 1 h–24 h: +++; more than 24 h: ++++.

**Table 3 ijerph-17-06506-t003:** Average content of D-limonene in some tree-derived essential oils [106].

Plant Source	Botanical Family	Part of the Plant Used	Content (%) *
*Boswellia rivae* Engl.	Burseraceae	Oleoresin	28.0–45.0
*Boswellia sacra* Flueck	Burseraceae	Oleoresin	6.0–21.9
*Bursera graveolens* (Kunth) Triana et Planch	Burseraceae	Oleoresin	58.6–63.3
*Canarium luzonicum* (Blume) A. Gray, *C. vulgare* Leenh.	Burseraceae	Oleoresin	26.9–65.0
*Ravensara aromatica* Sonnerat	Lauraceae	Leaves	13.9–22.5
*Abies alba* Mill.	Pinaceae	Leaves and branches	28.5–54.7
*Abies spectabilis* (D. Don) Spach	Pinaceae	Leaves and branches	29.6
*Pinus mugo* Turra	Pinaceae	Leaves and branches	6.1–37.1
*Citrus* spp.	Rutaceae	Fruit peel	27.0–95.0

* Content (%) = the relative quantity, expressed as a range of percentages, of D-limonene included in tree-derived essential oils.

**Table 4 ijerph-17-06506-t004:** Summary of findings of clinical studies and preclinical experiments with animal models about anti-proliferative effects of D-limonene.

Type	Doses	Results	Ref.
Animal model—mice	25 mg orally administered	Limonene reduces mice stomach/lung tumors by 3%	[112]
Animal model—rats	7.5–10% diet	Breast cancer regression in 89% of animals	[113]
Phase I—humans	0.5–12 g/m^2^	Partial response in breast cancer patients Absence of progression in subjects with colorectal cancer	[114]
Phase II—humans	8 g/m^2^	No responses in patients with solid tumors	[114]
Open-label—humans	2 g/die	22% reduction in cyclin D1 in early-stage breast cancer	[115]

**Table 5 ijerph-17-06506-t005:** Preclinical evidence of limonene biological activity.

Functional Response	Model	Molecules/Mechanisms Involved in Targeted Pathways	Ref.
**Anti-inflammatory** **Antioxidant**	Murine raw 264.7 cell line	TNF-α, IL-1, IL-6	[119]
	Human chondrocytes	NF-kB, NO, iNOS, p38, JNK	[70]
	Human lens epithelial cells	ROS, CASP, MAPK, Bcl-2/Bax	[120]
	Human neuroblastoma cells	LC3, clonogenic capacity	[121]
	Fruit fly	NO	[84]
	BALB/c mice	NO	[122]
	BALB/c mice	Catalase, peroxidase	[123]
	BALB/c mice	IL-5, IL-13, MCP-1, TGF-β	[124]
	BALB/c mice	NF-kB, p38, JNK, ERK	[125]
	Wistar rats	NF-kB, COX-2, iNOS	[126]
	Swiss mice	IL-1β	[127]
	Sprague–Dawley rats	COX2, ERK, iNOS, MMP-2, MMP-9, PGE, TGF-β	[128]
	BALB/c mice	Apoptosis-related genes	[129]
	Swiss mice	Oxidative stress	[130]
**Anxiolytic** **Antidepressant** **Sedative**	ICR mice	Elevated plus maze test	[90]
	Mice	Sleeping time	[88]
	Swiss mice	MBT assay, anxiolytic effect	[131]
	Rats	Locomotion, dopamine level	[132]
	Rats	Immobility in forced swim test	[133]
	CUMS mice	Body weight, sucrose preference, mobility	[134]
	*Mus musculus* albino mice	Elevated Plus Maze (EPM) test	[135]
**Analgesic**	Swiss mice	Induced nociception	[136]
	ICR mice	Writhing test	[90]
	Rats	Mechanical hyperalgesia	[133]
	Swiss mice	Writhing test	[127]
	Swiss mice	Mechanical hyperalgesia, IL-1β, TNF-α	[137]

Abbreviations: CASP, caspase; COX-2, cyclooxygenase-2; ERK, extracellular signal-regulated kinase; ICR, imprinted control region; IL, interleukin; iNOS, inducible NO synthase; JNK, c-jun N-terminal kinase; LC3, microtubule-associated protein light chain 3; MAPK, mitogen-activated protein kinase; MBT, marble burying test; MCP, monocyte chemoattractant protein; NF-kB, nuclear factor-B; NO, nitric oxide; p38, protein 38; PGE, prostaglandin E; ROS, reactive oxygen species; TGF, tumor growth factor; TNF, tumor necrosis factor.

**Table 6 ijerph-17-06506-t006:** Tree species whose total essential oil contains more than 20% of α-pinene [106].

Plant Source	Botanical Family	Part of the Plant Used	Content (%) *
*Boswellia frereana* Birdwood	Burseraceae	Oleoresin	41.7–80.0
*Boswellia sacra* Flueck	Burseraceae	Oleoresin	10.3–51.3
*Cupressus sempervirens* L.	Cupressaceae	Leaves	20.4–52.7
*Juniperus communis* L.	Cupressaceae	Fruit	24.1–55.4
*Juniperus phoenicea* L.	Cupressaceae	Leaves and branches	41.8–53.5
*Dryobalanops aromatica* Gaertn	Dipterocarpaceae	Wood	54.3
*Abies alba* Mill.	Pinaceae	Cones	18.0–31.7
*Larix laricina* Du Roi	Pinaceae	Leaves and branches	38.5
*Picea abies* (Mill.) Britton	Pinaceae	Leaves	14.2–21.5
*Pinus divaricata* Aiton	Pinaceae	Leaves and branches	23.1–32.1
*Pinus mugo* Turra	Pinaceae	Leaves	4.1–31.5
*Pinus nigra* J. F. X Arnold	Pinaceae	Leaves	11.5–35.1
*Pinus resinosa* Ait.	Pinaceae	Leaves and branches	47.7–52.8
*Pinus strobus* L	Pinaceae	Leaves	30.8–36.8
*Pinus sylvestris* L	Pinaceae	Leaves	20.3–45.8

* Content (%) = the relative quantity, expressed as a range of percentages, of α-pinene included in tree-derived essential oils.

**Table 7 ijerph-17-06506-t007:** Tree species whose total essential oil contains more than 20% of β-pinene [106].

Plant Source	Botanical Family	Part of the Plant Used	Content (%) *
*Abies alba Mill.*	Pinaceae	Cones	3.0–22.5
*Abies balsamea* L.	Pinaceae	Leaves and twigs	28.1–56.1
*Picea abies* (Mill.) Britton	Pinaceae	Leaves and twigs	4.8–31.9
*Picea glauca* (Moench) Voss	Pinaceae	Leaves and twigs	23.0
*Pinus mugo* Turra	Pinaceae	Leaves and twigs	1.3–20.7
*Pinus ponderosa* Douglas ex P. Lawson & C. Lawson	Pinaceae	Leaves and twigs	28.9
*Pinus resinosa* Ait.	Pinaceae	Leaves and twigs	29.4–29.9
*Pinus strobus* L.	Pinaceae	Leaves and twigs	31.1–33.3
*Pinus sylvestris* L.	Pinaceae	Leaves and twigs	1.9–33.3
*Tsuga canadensis* (L.) Carriere	Pinaceae	Leaves and twigs	20.8
*Citrus x aurantifolia Christm.*	Rutaceae	Fruit peel	21.1

* Content (%) = the relative quantity, expressed as a range of percentages, of β-pinene included in tree-derived essential oils.

**Table 8 ijerph-17-06506-t008:** Preclinical evidence of α-pinene biological activity.

	Model	Target	Reference
**Anti-inflammatory** **Antioxidant**	Murine macrophages	NF-kB, ERK, JNK	[152]
	Human U373-MG cells	ROS, peroxidase	[80]
	Human chondrocytes	NF-kB, IL-1β, JNK, iNOS, MMP-1, MMP-13	[153]
	Human lymphocytes	Total antioxidant capacity	[154]
	Mouse	Ig-E, IL-4	[149]
	Wistar rats	Superoxide dismutase, catalase, peroxidase, NO, IL-6	[155]
	C57BL/6 mice	CD4, CD8 and NK cells	[102]
	Wistar rats	COX2	[156]
**Anxiolytic** **Antidepressant** **Sedative**	Sprague–Dawley rats	Sleep rhythm	[157]
	Mice	BDNF, TH, EPM test	[158]
	ICR and C57BL/6N mice	GABA BZD receptor, sleep behavior	[159]
	Wistar–Kyoto mice	Forced swim test, oxidative phosphorylation expression	[160]
	Wistar rats	Sensorimotor severity score	[155]
	C57BL/6 mice	Schizophrenia-like behavior	[161]
**Analgesic**	BALB/c mice	Tail-flick test	[162]
	Mice	Neuropathic pain	[163]
	Wistar rats	Nociception	[156]

Abbreviations: BDNF, brain-derived neurotrophic factor; BZD, benzodiazepine; EPM, elevated plus maze; ERK, extracellular signal-regulated kinase; GABA, γ-aminobutyric acid; Ig-E, immunoglobulin E; JNK, c-jun N-terminal kinase; MMP-1, metalloproteinase 1; MMP-13, metalloproteinase 13; NF-kB, nuclear factor-B; ROS, reactive oxygen species; TH, tyrosine hydroxylase.

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
