# Peer review of "Forest Volatile Organic Compounds and Their Effects on Human Health: A State-of-the-Art Review"

_ijerph, 2020, doi:10.3390/ijerph17186506_

Round 1
Reviewer 1 Report
The review is topical, explores the connection between BVOCs produced in forest with potential health benefits for humans, and critically examines much of the evidence. The introduction is useful as it explains the terminology and places the review in the broader context of the environment (biotic, abiotic, physiology, physicochemical factors, etc.) and human well-being. Since the review title refers to forests, I would have expected some mention of BVOCs emanating from other components of the forests other than trees (e.g. soils, fungi, microbes, insects, herbs, see for example Laothawornkitkul et al. 2009 Biogenic volatile organic compounds in the Earth system. New Phytol. 183, 27–51 and your reference 19). Although the focus of the review is stated to be on pinene and limonene, it is quite wide ranging in the topics that are covered. I question whether the section on biochemistry of forest VOCs is necessary for the paper as it has been dealt with adequately in a number of other recent papers. Since the review focus is on pinene and limonene, Figure 2 should be redrawn to focus on the biosynthetic pathway, location in planta and storage of these compounds. The present figure is quite poor – a more interesting style figure is Figure 2 in Šimpraga et al 2019 (reference 19). The coverage of distillation products and their use in Section 3.2 A qualitative analysis of forest VOCs emissions is out of place – it warrants a separate section heading and should be placed later in the MS. This would allow all the relevant information on essential oils products to be integrated into one section. The jumping back and forth between VOCs in forests and laboratory products as essential oils in distracting from the main thrust of the review. In place of levels in tree-derived essential oils (Table 3), please provide data for contents in trees of interest and forest atmospheric concentrations. These are more relevant than steam distilled products. The Discussion though generally informative would benefit from several modifications. First, please discuss how relevant the doses used in clinical studies are to concentrations measured in forests, and whether this limits how confident we can be on the assumed direct specific health benefits of forest VOCs in nature. Where are the gaps in knowledge and what environmental studies should be undertaken. Second, avoid extending the discussion beyond the boundaries of the review - section 4.3 is rather discursive and wanders from the topic – shorten and deal directly with the topic of this review. Other points to consider when revising the MS are: Why are the terpene producing oil glands in leaves of many evergreen forests largely ignored? The first two sentences of 3.3.1 require a reference. What does the content (%) in Table 6 refer to (is it oil composition or is it based on the mass of plant material that was extracted)? Consider stating the number of references that were identified from the data bases and then the number that were found to be useful in the review.Author Response
First, we would like to thank the editors and the referees for evaluating our article and providing us with valuable feedback to improve the manuscript. All raised issues have been addressed point by point, as stated below. Changes have been highlighted in the manuscript. Here, our replies have been written in bold letters for better reading.
REVIEWER #1
The review is topical, explores the connection between BVOCs produced in forest with potential health benefits for humans, and critically examines much of the evidence. The introduction is useful as it explains the terminology and places the review in the broader context of the environment (biotic, abiotic, physiology, physicochemical factors, etc.) and human well-being.
Thanks for your kind appreciation of our research work.
Since the review title refers to forests, I would have expected some mention of BVOCs emanating from other components of the forests other than trees (e.g. soils, fungi, microbes, insects, herbs, see for example Laothawornkitkul et al. 2009 Biogenic volatile organic compounds in the Earth system. New Phytol. 183, 27–51 and your reference 19).
As requested by the reviewer, an entire section of the Introduction (numbered 1.2) has been dedicated to this interesting topic, thus providing the review with a brief overview on the importance of non-tree-derived BVOCs, which can play a more indirect role in the composition of the forest air.
Although the focus of the review is stated to be on pinene and limonene, it is quite
wide ranging in the topics that are covered. I question whether the section on biochemistry of forest VOCs is necessary for the paper as it has been dealt with adequately in a number of other recent papers.
We have decided to keep our detailed introductory section about biochemistry as is in order to improve the comprehensiveness of our research work, thus turning our article into a real state-of-the-art review stretching from basic science to medicine and public health. The other reviewer has highly appreciated this approach and we agree with their position.
Since the review focus is on pinene and limonene, Figure 2 should be redrawn to focus on the biosynthetic pathway, location in plants and storage of these compounds. The present figure is quite poor – a more interesting style figure is Figure 2 in Šimpraga et al 2019 (reference 19).
Following the reviewer’s suggestion, it was decided to improve the quality and readability of Figure 2. Two graphs (Figures 3 and 4) have been also added to display the average concentration of the most relevant BVOCs in the forest air (as requested by the reviewer in the next comment). Additionally, a graphical abstract has been created to visually synthetize the main findings of our review.
The coverage of distillation products and their use in Section 3.2 A qualitative analysis of forest VOCs emissions is out of place – it warrants a separate section heading and should be placed later in the MS. This would allow all the relevant information on essential oils products to be integrated into one section. The jumping back and forth between VOCs in forests and laboratory products as essential oils in distracting from the main thrust of the review. In place of levels in tree-derived essential oils (Table 3), please provide data for contents in trees of interest and forest atmospheric concentrations. These are more relevant than steam distilled products.
It was decided to separate the qualitative analysis of forest VOCs from the paragraph describing the properties of essential oils, thus creating two different sections (3.2 and 3.3). The paragraph describing the analysis was not placed later in the manuscript because it was reputed necessary to follow this order in describing the Results of our review: biochemistry of VOCs (Figure 2), qualitative analysis (Table 2), quantitative analysis (Figures 3 and 4), properties of essential oils with a substantial content of forest VOCs, properties of VOCs, effects of limonene and pinene on human health.
Table 3 has been kept in the manuscript, but, following the reviewer's recommendations, two dedicated graphs (Figures 3 and 4) along with their description have been added to visually display the average concentration of the most relevant BVOCs in the forest air.
The Discussion though generally informative would benefit from several modifications.
First, please discuss how relevant the doses used in clinical studies are to concentrations measured in forests, and whether this limits how confident we can be on the assumed direct specific health benefits of forest VOCs in nature. Where are the gaps in knowledge and what environmental studies should be undertaken.
The first part of the Discussion (Section 4.1) has been markedly expanded and improved, thus adding more information about the differences in BVOCs pharmacokinetics when taken orally in experimental settings and when inhaled during a forest trip. Gaps in knowledge and possible suggestions for future studies have been also discussed more in depth.
Second, avoid extending the discussion beyond the boundaries of the review - section 4.3 is rather discursive and wanders from the topic – shorten and deal directly with the topic of this review.
As requested, Section 4.3 of the Discussion has been shortened to better keep the focus on the primary aim of our research.
Other points to consider when revising the MS are: Why are the terpene producing oil glands in leaves of many evergreen forests largely ignored?
While a more specific description of the detailed origin of plant-derived BVOCs would be indeed interesting, as it would help to identify environmental parameters which can influence the release of BVOCs and their flux variations, this in-depth analysis is not directly relevant to our main objective. However, in the introduction of Section 1.2 we have mentioned that different plant parts can emit BVOCs.
The first two sentences of 3.3.1 require a reference.
The proper reference has been added and the two sentences rephrased for better reading.
What does the content (%) in Table 6 refer to (is it oil composition or is it based on the mass of plant material that was extracted)?
The percentage refers to essential oil composition. This detail has been now specified in the legends of Tables 6 and 7.
Consider stating the number of references that were identified from the data bases and then the number that were found to be useful in the review.
This information has been added to the methodological section of the article.
Reviewer 2 Report
The paper by Antonelli et al. is a remarkable and comprehensive review that introduces readers to the complex world of volatile organic compounds (VOCs) released from trees. The review leads the reader through the basic chemistry, biological diversity, and hypothesized human-health applications of VOCs. Given the wide a varied interest, especially from the Nature as Medicine community, in the role of VOCs in psychological and physiological restoration in response to engagement with nature, this paper should be considered essential reading.
I have a few suggestions that might improve the clarity of the manuscript.
- The abstract does not give the reader an indication of the in depth presentation of the chemistry and diversity of VOCs in the paper. This is an important component of the paper and should be in the abstract.
- Use "diurnal rhythm" instead of "circadian rhythm". The term circadian is defined to mean that the rhythm is generated by an underlying endogenous pacemaker mechanism and that the rhythm will free-run when isolated from time-giving cues (zeitgebers). Diurnal rhythms are rhythms that occur on a 24 hour periodicity, but is not necessarily the product of an endogenous biological clock. https://en.wikipedia.org/wiki/Circadian_rhythm.
- Figure 1 is confusing and hard to read. The font is too small. It is really not clear what relationships the arrows are intended to imply. Please, reconsider the purpose of this diagram.
- Table 2. The formatting of the column headings is disruptive. Try reformatting to reduce the truncation and division of words. The last two column headings (Emissions and Persistence) need a subscript to indicate they are defined in a footnote at the end of the lengthy table. In the draft manuscript that I read, the footnote containing the definition appears at the top of page 13 and does not seem to be associated with the table. This made page 13 very confusing until I figured out the relationship to the table.
- Page 13 ("... GLVs (although rarely...). "Rarely" seems to be the wrong word, I think "occasionally" might be a better word choice.
- Page 15, first line. After carefully distinguishing "phytoncide" from "VOCs", the authors revert to the less specific term phytoncide. Is there a reason for this reversion? If not, I would encourage them to continue to use the specific and accurate terminology they have employed in the rest of the manuscript.
- Table 2. "Content &", please explain what the "percentage" is relative to - volume? dried plant mass? etc.
- Page 15, introduction to section 3.4.1 they describe a variety of pro- and anti-inflammatory effects, but make no reference as to the type of systems in which the responses were observed. To interpret the results it is important to know if these are in vivo (and, if so, what types of organisms, not just "animal models", be specific) or in vitro (and, if so, what cell types).
- Table 4 lists "animal model" - what types of animals models?
- Table 5. Adding a column heading for the furthest left column would be helpful. Perhaps "Functional Response" would serve? Also, it is unclear what is meant by "Target".
- Section 3.4.5. "...PGE2 production [21]. In in vitro...". Although I understand it is difficult if not impossible to give details of all the cell types used, providing some indication of the cell types used would be helpful.
- Section 3.5. What is meant by "most important tree species"? For example, is this in terms of the abundance of trees in forests? The volume of VOCs produced? Please, clarify.
- Table 6. "More than 20%" -what is the base of this calculation? 20% of what?
- Table 7. I am unfamiliar with the term "leavers", is this supposed to be "leaves"?
- "Balsamic effects", this is not a familiar term and a quick Google search only turned up references to balsamic vinegar. Please, define.
- Table 9 is a wonderfully clear and concise summary of compounds and their effects.
- The Discussion is a thoughtful and well-integrated presentation of the field.
- References. Check formatting of references for consistency. The title of reference 2 is in all capitol letters
Author Response
First, we would like to thank the editors and the referees for evaluating our article and providing us with valuable feedback to improve the manuscript. All raised issues have been addressed point by point, as stated below. Changes have been highlighted in the manuscript. Here, our replies have been written in bold letters for better reading.
REVIEWER #2
The paper by Antonelli et al. is a remarkable and comprehensive review that introduces readers to the complex world of volatile organic compounds (VOCs) released from trees. The review leads the reader through the basic chemistry, biological diversity, and hypothesized human-health applications of VOCs. Given the wide a varied interest, especially from the Nature as Medicine community, in the role of VOCs in psychological and
physiological restoration in response to engagement with nature, this paper should be considered essential reading.
Thanks for your kind appreciation of our research work.
I have a few suggestions that might improve the clarity of the manuscript.
- The abstract does not give the reader an indication of the in depth presentation of the chemistry and diversity of VOCs in the paper. This is an important component of the paper and should be in the abstract.
The fact that our paper includes a thorough analysis of the chemistry and diversity of forest VOCs has been now reported in the abstract.
- Use "diurnal rhythm" instead of "circadian rhythm". The term circadian is defined to mean that the rhythm is generated by an underlying endogenous pacemaker mechanism and that the rhythm will free-run when isolated from time-giving cues (zeitgebers). Diurnal rhythms are rhythms that occur on a 24 hour periodicity, but are not necessarily the product of an endogenous biological clock.https://en.wikipedia.org/wiki/Circadian rhythm.
Following the reviewer’s suggestion, the term “circadian rhythm” has been changed with “diurnal rhythm” in the abstract and in paragraphs 3.2 and 4.1.
- Figure 1 is confusing and hard to read. The font is too small. It is really not clear what relationships the arrows are intended to imply. Please, reconsider the purpose of this diagram.
As requested, the quality and design of Figure 1 have been substantially improved. Additionally, a graphical abstract has been created to visually synthetize the main findings of our review.
- Table 2. The formatting of the column headings is disruptive. Try reformatting to reduce the truncation and division of words. The last two column headings (Emissions and Persistence) need a subscript to indicate they are defined in a footnote at the end of the lengthy table. In the draft manuscript that I read, the footnote containing the definition appears at the top of page 13 and does not seem to be associated with the table. This made page 13 very confusing until I figured out the relationship to the table.
Table 2 has been now formatted in such a way as to have as few truncated words as possible in order to improve its readability. The table legend has been expanded, rewritten and superscript numbers have been added to create clear links to column headings.
- Page 13 ("... GLVs (although rarely...). "Rarely" seems to be the wrong word, I think "occasionally" might be a better word choice.
Following the reviewer’s suggestion, the term “rarely” has been changed with the more proper “occasionally” in page 13.
- Page 15, first line. After carefully distinguishing "phytoncide" from "VOCs", the authors revert to the less specific term phytoncide. Is there a reason for this reversion? If not, I would encourage them to continue to use the specific and accurate terminology they have employed in the rest of the manuscript.
Thanks for reporting this information. We have now aligned the entire document to the same terminology, thus only employing the term “VOCs” to avoid misunderstandings.
- Table 2. "Content &", please explain what the "percentage" is relative to - volume? dried plant mass? etc.
Probably, the reviewer refers to Table 3 instead of Table 2. This detail has been now provided in the table legends.
- Page 15, introduction to section 3.4.1 they describe a variety of pro- and anti-inflammatory effects, but make no reference as to the type of systems in which the responses were observed. To interpret the results it is important to know if these are in vivo (and, if so, what types of organisms, not just "animal models", be specific) or in vitro (and, if so, what cell types).
This detail has been now added to the introduction of section 3.4.1.
- Table 4 lists "animal model" - what types of animal models?
The type of animal models has been specified in Table 4.
- Table 5. Adding a column heading for the furthest left column would be helpful. Perhaps "Functional Response" would serve? Also, it is unclear what is meant by "Target".
The first column heading has been added as recommended by the reviewer. The heading of the third column has been modified for better clarity.
- Section 3.4.5. "...PGE2 production [21]. In in vitro...". Although I understand it is difficult if not impossible to give details of all the cell types used, providing some indication of the cell types used would be helpful.
Cell types used for in vitro experiments have been mentioned in section 3.4.5, as requested by the reviewer.
- Section 3.5. What is meant by "most important tree species"? For example, is this in terms of the abundance of trees in forests? The volume of VOCs produced? Please, clarify.
In order to better explain this detail, the expression “important tree species” has been changed with “common tree species”. Here, we meant to refer to tree types more commonly found in green areas and forests.
- Table 6. "More than 20%" -what is the base of this calculation? 20% of what?
This detail has been now specified in Tables 6 and 7 (it refers to the relative quantity of pinene over the total essential oil of specific trees).
- Table 7. I am unfamiliar with the term "leavers", is this supposed to be "leaves"?
I confirm that this was a typo (we meant “leaves”). Thanks for reporting it.
- "Balsamic effects", this is not a familiar term and a quick Google search only turned up references to balsamic vinegar. Please, define.
This is probably due to our imprecise use of an Italian-derived word. “Balsamico'', in our native language, refers to any substance with anti-inflammatory, antioxidant and protective effects on a mucosal or skin surface. In order to avoid misunderstandings, the term “balsamic” has been erased, thus only leaving more proper adjectives (“antioxidant and anti-inflammatory”) in the abstract, in section 4.1 and in the conclusions.
- Table 9 is a wonderfully clear and concise summary of compounds and their effects.
Thanks for the appreciation of this part of our research work.
- The Discussion is a thoughtful and well-integrated presentation of the field.
Thanks for the appreciation of this part of our research work.
- References. Check formatting of references for consistency. The title of reference 2 is in all capital letters
Bibliography has been checked and the title of reference 2 has been rewritten correctly.
Round 2
Reviewer 1 Report
The authors have undertaken the key recommendations and this has resulted in a much stronger review.
I suggest that the information in Table 7 be reordered alphabetically by taxa within families
Author Response
Thanks for your feedback. As requested, information in all tables where plant sources are mentioned (including Table 7) have been reordered alphabetically by taxa within families.